# Short-Term Panax Ginseng Extract Supplementation Reduces Fasting Blood Triacylglycerides and Oxygen Consumption during Sub-Maximal Aerobic Exercise in Male Recreational Athletes

**DOI:** 10.3390/biom14050533

**Published:** 2024-04-30

**Authors:** Didier Hernández-García, Ana Belén Granado-Serrano, Meritxell Martín-Gari, Assumpta Ensenyat, Alba Naudí, Jose C. E. Serrano

**Affiliations:** 1Department of Experimental Medicine, NUTREN-Nutrigenomics, Universitat de Lleida, 25198 Lleida, Spain; didier.hernandez@udl.cat (D.H.-G.); anabgs@gmail.com (A.B.G.-S.); meritxell.martin@udl.cat (M.M.-G.); albanaudi@gmail.com (A.N.); 2Institut Nacional d’Educació Física de Catalunya, 08038 Lleida, Spain; aensenat@inefc.udl.cat

**Keywords:** sub-maximal aerobic capacity, endurance, blood lipids, *Panax ginseng*, recreational athletes

## Abstract

Ginseng, a popular herbal supplement among athletes, is believed to enhance exercise capacity and performance. This study investigated the short-term effects of Panax ginseng extract (PG) on aerobic capacity, lipid profile, and cytokines. In a 14-day randomized, double-blind trial, male participants took 500 mg of PG daily. Two experiments were conducted: one in 10 km races (*n* = 31) and another in a laboratory-controlled aerobic capacity test (*n* = 20). Blood lipid and cytokine profile, ventilation, oxygen consumption, hemodynamic and fatigue parameters, and race time were evaluated. PG supplementation led to reduced total blood lipid levels, particularly in triacylglycerides (10 km races −7.5 mg/dL (95% CI −42 to 28); sub-maximal aerobic test −14.2 mg/dL (95% CI −52 to 23)), while post-exercise blood IL-10 levels were increased (10 km 34.0 pg/mL (95% CI −2.1 to 70.1); sub-maximal aerobic test 4.1 pg/mL (95% CI −2.8 to 11.0)), and oxygen consumption decreased during the sub-maximal aerobic test (VO_2_: −1.4 mL/min/kg (95% CI −5.8 to −0.6)). No significant differences were noted in race time, hemodynamic, or fatigue parameters. Overall, PG supplementation for 2 weeks showed benefits in blood lipid profile and energy consumption during exercise among recreational athletes. This suggests a potential role for PG in enhancing exercise performance and metabolic health in this population.

## 1. Introduction

*Panax ginseng,* C.A. Meyer (PG), also known as Red, Chinese, or Korean ginseng, is a plant native to eastern Asia. Its root contains over 30 saponin components known for their diverse biological activities, such as immune restoration, anti-fatigue properties, and neurovegetative effects [1]. For these reasons, ginseng is one of the most popular herbal supplements in the world used as an aid to treat and prevent many ailments, but also to increase occupational efficiency involving physical work and increase physical stamina [2]. Accumulative evidence has shown that prolonged use of ginseng or its active components affects behavior and psychomotor and exercise performance [3], and recent surveys indicate that ginseng is one of the most popular herbal supplements consumed by elite athletes [4,5,6]. 

Several potential targets have been identified to explain the suggested properties during physical activity. For example, it has been observed in animal models that ginseng supplementation may protect muscles from eccentric exercise injuries, preserving mitochondrial membrane integrity and reducing carbonyl and nitrate muscle content [7]. Similarly, treatment with PG extract G115 increases the capillary density and the oxidative capacity of muscles with greater aerobic potential in exercised rats [8], and may promote fat oxidation and spare muscle and hepatic glycogen content during exercise [9]. 

Notwithstanding, placebo-controlled studies examining the ability of ginseng to improve endurance performance in humans are limited. Some studies suggest that ginseng supplementation may enhance maximum oxygen consumption (VO_2max_) and anaerobic power [10] and recently these ergogenic effects have been attributed to the ginsenoside content [11]. Some of these effects could be explained due to the ability of ginseng to change blood metabolic profile, in pathways involved in lipid metabolism, energy balance, and chemical signaling [12], indicating that the alteration in metabolic profiling may reflect the anti-fatigue impact of ginseng.

Nonetheless, other studies do not support any ergogenic effect on peak aerobic exercise performance [13] and lactate threshold [14]. The absence of compelling research demonstrating the ability of ginseng to consistently enhance physical performance in humans may be due to the variability of the supplement and/or different methods of study employed. Additionally, there is a lack of information regarding the effects of ginseng supplementation in real-life conditions and/or during sub-maximal aerobic conditions. There is some evidence that PG can improve physical performance through an improvement in lipid metabolism. However, few studies have evaluated the effects of PG supplementation in sub-maximal physical activity where lipid metabolism is a key factor. Therefore, the purpose of this study was to determine the short-term (14-day) effect of daily supplementation of 500 mg of PG extract vs. placebo on sub-maximal aerobic exercise and lipid metabolism in recreational athletes. The current study hypothesis is that reported modifications in lipid metabolism induced by PG could serve as an ergogenic aid during exercise at sub-maximal aerobic exercise performance. 

## 2. Materials and Methods

### 2.1. Experimental Design

Two randomized, double-blind, placebo-controlled trials with different volunteers in each trial were conducted. The first experiment was performed in real-life conditions, in which recreational athletes (*n* = 31) were supplemented with PG extract or placebo, and the outcomes determined were changes in blood lipid profile, race time, and fatigue perception after participation in two 10 km races. The second experiment was performed in another cohort of recreational athletes (*n* = 20) with the same characteristics as the first experiment. It was performed in laboratory-controlled conditions, with the same supplementation protocol, and the outcomes determined were changes in blood lipid profile, aerobic capacity variables, and fatigue perception after a sub-maximal aerobic exercise performance test. 

#### 2.1.1. First Experiment: 10 km Races

Participants completed two 10 km races located around the city of Lleida, the first race on 16 December 2017, and the second race on 31 December 2017. Both races were homologated in distance and accumulated slope and were performed within two weeks, which corresponded to the two weeks of treatment duration. Blood samples were obtained one day before the race in fasting conditions and at each race immediately after the volunteer crossed the finish line. Part of the research team directed the volunteer to the station that the research group had near the finish line, and capillary blood was obtained within a maximum time of 5 min. Organization electronic chips recorded the race time, and the general feeling of perceived fatigue was determined by Borg’s rating of perceived exertion CR-10 scale immediately after the end of both races. 

#### 2.1.2. Second Experiment: Sub-Maximal Aerobic Test

Sub-maximal aerobic exercise was defined as the velocity in which the respiratory exchange ratio (RER) is 0.94, which corresponds to 70% of lipids’ energy source during exercise [15]. A maximal aerobic capacity test was performed two weeks before the study intervention to determine the volunteer’s velocity at 0.94 RER. The maximal aerobic capacity test was performed on a treadmill with a slope of 3% and all volunteers completed a previous ad libitum warm-up session for 5 min. The test started with a 6 km/h velocity, maintaining this speed for 3 min, followed by 1 km/h velocity increments every minute until the exhaustion of the athlete. O_2_ consumption and CO_2_ production were measured with an Ergostik Geratherm (Blue Cherry, Bad Kissingen, Germany) indirect calorimeter. The aerobic and anaerobic thresholds and the maximum aerobic capacity (VO_2max_) were determined and, further, the running speed at which the RER corresponds to 0.94 was established as the point of sub-maximal aerobic exercise. 

Further, two sub-maximal aerobic exercise tests were performed, one before the supplementation period and the second after 14 days of PG extract/placebo supplementation. Participants performed the same warm-up protocol and velocity increments until reaching individual sub-maximal aerobic velocity, and this velocity was maintained for 20 min (sub-maximal aerobic period), after which volunteers stopped running and were asked to sit up for 3 min (resting period). Figure 1 describes the scheme of the followed protocol. O_2_ consumption, CO_2_ production, and cardiopulmonary variables were measured during the whole test. Blood samples were drawn before and 10 min after the test was ended. Additionally, blood lactate levels were quantified before starting the test, 10 min after starting the sub-maximal aerobic period, at the end, and 10 min after the end of the test with a portable lactate reader (Accutrend Plus, Roche, Mannheim-Germany). Finally, effort perception was asked by Borg’s scale during the sub-maximal aerobic period at minutes 3, 12, and 19. 

### 2.2. Participants

The inclusion criteria in both experiments were: male recreational athletes, ages older than 18 years old, with stable training workout and frequency equal to or superior to 3 times per week, and no planned modifications in training parameters during the study period time. The exclusion criteria were athletes with high blood pressure, pharmacological treatment for hypercholesterolemia, diabetes, psychiatric illness, and/or any medication that may modify lipid metabolism, and the use of any type of ergogenic supplementation in the previous 3 months of the study period time. 

The sample size (α-error: 0.05, β-error: 0.10) was calculated based on the expected outcomes due to PG extract supplementation of an improvement in running time by 60 s (standard deviation of 60 s) and a change in 3 mL/Kg/min in VO_2_ (standard deviation of 4.7 mL/Kg/min) for the first and second experiment, respectively. The minimum number of volunteers in each experiment was 16 and 10 volunteers per group in the 10 km race and sub-maximal aerobic test, respectively. The recruitment flow diagram for both experiments is described in Figure 2. Participants were randomized through the EPIDAT computer program, separating them into two groups: PG and placebo groups. Double blinding was conducted for all study participants and researchers. Participant demographic, training, and nutritional characteristics for both experiments are described in Table 1. 

Both studies were conducted following the principles of the Declaration of Helsinki and Good Clinical Practice guidelines. The study protocol registration number is NCT03888196 and was approved by the University Hospital Arnau de Vilanova Institutional Ethics Committee (approval number CEIC-1742). All participants provided written informed consent. 

### 2.3. Interventions

The intervention in both experiments consisted of a daily dose, in a single capsule, of 500 mg of a dry extract of Panax ginseng (PG group) or 500 mg of microcrystalline cellulose (placebo group) taken in fasting conditions or two hours before physical activity in training days. The intervention time was 14 days. In both cases, it was suggested to take the capsule with a glass of water at least 30 min before the ingestion of any type of food. PG extract and microcrystalline cellulose were purchased from A.C.E.F. spa (Fiorenzuola D’Arda, Piacenza, Italy) and packaged in green-color gelatine capsules. PG root extract was of Chinese origin and was extracted with water:ethanol (50:50) containing at least 30% of ginsenosides (Lot number N20626L0 and certificate of analysis No. 85341). Dosage and treatment time were determined based on the results expressed in a systematic review evaluating the efficacy and safety of ginseng [16]. Few clinical trials have been performed on physical performance, most of them with exposure time between 1 and 8 weeks. The rationale for the treatment dosage was established on the premise that effects on physical activity might be observed at higher doses than those used in previous studies (between 200 and 400 mg/day of PG extract). Due to the high doses, it was deemed appropriate for the exposure time to be short, avoiding potential adverse effects of PG overexposure. The adherence to the treatment was controlled by capsules blister collection after the end of the treatment period, and a weekly training diary supervised workout. Similarly, adverse effects were registered during the treatment period.

### 2.4. Measurements

#### 2.4.1. Energy Cost

Energy cost was estimated in the sub-maximal aerobic exercise test. It is defined as the quantity of energy spent per distance unit (expressed in mL O_2_/kg/km), and was calculated by the di Prampero formula (Energy cost = (VO_2_ During Test-VO_2_ During Rest)/speed) [17]. 

#### 2.4.2. Blood Biochemical Analysis

Blood biochemical profile was determined by enzymatic spectrophotometric assays from Spinreact (St. Esteve de Bas, Spain) following the manufacturer’s instructions with the following analysis kits: lactate (1001330); total-cholesterol (1001095); triacylglycerides (1001312); phospholipids (1001140); and total lipids (1001270). Non-esterified fatty acids (NEFA) were determined by a colorimetric method previously described [18], and glycerol by enzymatic fluorometric assay kit (MAK117) from Sigma-Aldrich (St. Louis, MO, USA). 

#### 2.4.3. Ex Vivo Cellular Respiratory Capacity

Volunteer plasma samples after the intervention period were exposed to a muscle cell culture for 24 h to determine modifications in cellular respiratory capacity. Briefly, immortalized rat skeletal (L6 cell line) myoblast cells were maintained in Dulbecco’s modified Eagle’s medium supplemented with 10% fetal bovine serum at 37 °C in a 5% CO_2_ atmosphere. Once the myoblasts were grown to confluence, the medium was replaced with Dulbecco’s modified Eagle’s medium containing 2% fetal bovine serum to induce differentiation into myotubes. After that, 2% fetal bovine serum was substituted with volunteer plasma and incubated for 24 h at 37 ºC in a 5% CO_2_ atmosphere. Routine respiration was studied in duplicate in a respirometry chamber (Oroboros Instruments, Innsbruck, Austria) [19] containing 2 mL MIR05 (sucrose 110 mM; potassium lactobionate 60 mM; EGTA 0.5 mM; MgCl_2_·6H_2_O 3 mM; taurine 20 mM; KH_2_PO_4_ 10 mM; Hepes 20 mM; BSA,1 g/L; pH 7.1 at 37 °C). Further, uncoupled respiration was determined by the addition of 0.5 μM of Carbonyl cyanide 4-(trifluoromethoxy)phenylhydrazone. Spare respiratory capacity was determined by the difference between uncoupled respiration and routine respiration.

### 2.5. Nutrition and Exercise Training Parameters

Dietary intake was assessed through dietary recall on the day of the 10 km race or the sub-maximal aerobic test. Data were collected by qualified dietitians and subsequently analyzed using the DIAL software (Version 1.04.02, Alce Ingenieria, Madrid, Spain), which estimates nutrient intake based on a database of Spanish foods. Training records were obtained through a weekly form that included the date, type of training, distance covered, and duration. All forms were collected prior to the tests being conducted.

### 2.6. Statistical Analysis

Data are presented as mean (lower-upper 95% confidence interval) or mean ± standard error of the mean in data where replicate analysis was performed. Analysis of the Gaussian distribution of all variables was performed by the D’Agostino–Pearson omnibus normality test. Outlier analysis was performed, and no data were excluded, in all analyses. The statistical significance comparison of variables with a normal Gaussian distribution was calculated by a paired *t*-test, two-way ANOVA analysis with LSD as a post hoc test, or Pearson correlation analysis. A non-normal Gaussian distribution was analyzed by a Wilcoxon matched-paired signed-rank test, Kruskal–Wallis test with Dunn’s as a post hoc test, or Spearman correlation analysis. In both cases, a two-tailed value of *p* below 0.05 in all statistical tests employed was considered statistically significant. Cohen’s *d* was determined by calculating the mean difference between groups, and then dividing the result by the pooled standard deviation. GraphPad Prism (version 5.0) was used for statistical analysis and graph plotting. The results of the statistical analyses are provided in the Supplementary Information.

## 3. Results

The demographic characteristics of the volunteers who participated in the study are shown in Table 1. There were no significant differences in these parameters between volunteers before the experimental period began. However, at the end of the experimental period, it was observed that volunteers who underwent treatment with PG extract showed an increase in carbohydrates in both cohorts and protein intake before the sub-maximal aerobic test. There were no observed changes between both groups in the physical activity recorded by the volunteers. No adverse events were reported due to PG extract exposure in both experiments. The dropouts during the experimental period were mainly due to schedule conflicts, non-adherence to the treatment, sports injuries, and flu (Figure 1). 

The blood lipid profile of volunteers before and after the 10 km race in the PG vs. placebo groups is described in Table 2. No significant changes were observed in the lipid profile of the volunteers before and after the first race in both the PG and placebo groups. After two weeks of intervention, it was observed that the PG group presented a reduction in pre-race blood total lipids and triacylglycerides (Cohen’s *d* = 1.49 and 0.62, respectively), while no differences in phospholipids, total cholesterol, or non-esterified fatty acids were observed. No differences were noted in lipid levels post-race following PG treatment. From a broader perspective, the decline in total lipid and triacylglyceride levels induced by PG treatment did not correlate with changes in race time (placebo vs. PG time difference: 0.4 min [95% CI: 0.7 to 1.5]) or fatigue perception (placebo vs. PG Borg scale difference: 0.6 [−0.9 to 2.0, 95% CI]). Regarding blood cytokines, both groups exhibited an increase in IL-10 levels post-race. As a distinct effect of PG supplementation, elevated levels of IL-6 and IL-8 were observed following the 10 km race (Cohen’s *d* = 0.91 and 1.99, respectively) (Table 2). Regarding blood lactate levels, an increase in its levels was observed after both races. However, the PG group showed higher post-race levels of lactate compared to the control group (Cohen’s *d* = 7.49 and 5.43 for the first and second races, respectively). 

To determine if the observed lipid-lowering properties of ginseng may influence athletes’ performance, a sub-maximal test in controlled laboratory conditions was performed. The changes in lipid profile before and after the supplementation period are described in Table 3. Like what was observed in the 10 km races, after two weeks of PG supplementation, there was a reduction in blood total lipids and triacylglycerides levels without any modification in blood lipid levels after the sub-maximal test (Cohen’s *d* = 2.05 and 1.12, respectively). Regarding blood cytokines, few differences were observed before and after the sub-maximal performance aerobic test. The main differences were denoted by the higher increase in IL-1Ra and IL-10 levels after the test in the PG group (Cohen’s *d* = 0.78 and 1.08, respectively).

Regarding performance variables, Table 3 and Figure 3 summarize the changes in cardiopulmonary parameters obtained in the sub-maximal exercise test. No changes in RER were observed during the test; similarly, heart rate was maintained between 143 and 164 bpm, which corroborates that the exercise was performed at sub-maximal aerobic performance. As a main result, a decrease in oxygen consumption of 1.4 mL/min/kg [−5.8 to −0.6, 95% CI] (Cohen’s *d* = 0.47) in the volunteers treated with PG (Figure 3A) was observed. The difference in oxygen consumption was observed without showing significant differences in ventilation, respiratory coefficient, or heart rate (Figure 3B–D). Regarding fatigue parameters, it was observed that, between the first and second tests, the volunteers reported a higher perception of effort in both groups (0.5 points on the Borg’s scale), although these differences were not significant (Figure 3E). Similarly, Figure 3F shows the changes in blood lactate concentration before (time 0 min) and after (time 20 min) the sub-maximal aerobic test. It is appreciated that both groups present an average increase of 1.5 mg/dL in lactate without showing a difference in these levels between the groups. The changes in oxygen consumption reflected a significant reduction in energy cost during the sub-maximal aerobic test (Figure 3G). To determine if the reduction in oxygen consumption could be related to an increase in lipid metabolism, the difference in plasma glycerol levels after the sub-maximal exercise test was determined (Figure 3H). As a result, an increase in glycerol levels was observed (32 μM [8 to 56, 95% CI] for the placebo group and 39 μM [24 to 54, 95% CI] for the PG group), notwithstanding that no differences in the increase in glycerol levels were observed between groups. 

During the resting period, after the 20 min sub-maximal aerobic test, the main difference observed was an increase in heart rate in the volunteers treated with PG, which was 11.4 bpm (0.6 to 22.2, 95%CI) higher than the placebo group. No difference in VO_2_, ventilation or respiratory coefficient between both groups was observed in the resting period. 

Finally, the plasma of the volunteers was exposed to the L6 muscle cell line for 24 h and the respiration capacity was analyzed (Table 4). The plasma of the volunteers supplemented with PG induced a decrease in oxygen consumption in routine conditions of 151 pmol/s, as well as an increase in the spare respiratory capacity of 172 pmol/s. However, these changes did not reach the limit of significance proposed in this study. Contrarily, an increase in cellular lactate levels was observed after the protocol experiment.

## 4. Discussion

The main finding of this study is that a daily dose of 500 mg (30% of ginsenosides) of PG extract reduces oxygen consumption in a laboratory-controlled environment during sub-maximal aerobic exercise in recreational athletes. This finding may be explained by the potential increase in energy efficiency due to the observed changes in blood lipid profile and cytokine levels.

Previous studies have described contradictory results regarding the potential of PG as an ergogenic agent. Some of them show positive effects, while others do not have any effect on physical performance variables [20]. It was suggested that the lack of coincidence in the results might be attributed to the low standardization and low dose of ginsenosides in PG supplements employed [13]. Similarly, PG formulations with higher content of ginsenosides (above 10%) have shown improvements in VO_2max_ and muscular strength [11]. This may suggest that the higher ginsenoside content of the PG extract formulation employed in this study should be enough to induce changes as observed. Moreover, all previous studies determined the ergogenic capacity of PG measuring maximal physical performance through the determination of VO_2max_. In this study, it was hypothesized that the change in lipid metabolism induced by PG extract may exert benefits during sub-maximal anaerobic capacity where lipid metabolism plays a key role.

In this regard, it was observed that the reduction in oxygen consumption induced by PG extract supplementation might be partially explained by the changes in lipid homeostasis. Changes in lipid metabolism by PG have been reported elsewhere; in a recent meta-analysis, it was suggested that PG may improve lipid profile mainly through a reduction in total and LDL-cholesterol [21]. Results from this study suggest that PG extract supplementation before exercise in recreational subjects may modify lipid homeostasis due to the observed reduction in fasting TG levels in both cohorts. Nevertheless, with the information obtained from this study, it is not possible to determine if the reduction in TG levels can be attributed to increased use of lipids as a source of energy during exercise. In normal physiology, fat oxidation is highest during moderate-intensity endurance exercise [22]; furthermore, fat oxidation is elevated in the post-exercise state. Although very-low-density lipoproteins and chylomicrons contribute to muscle substrate supply, there is no support for increased TG output from the liver upon acute exercise [23], which indicates that the lower fasting levels of TG observed could be attributable to a clearing effect of PG after exercise. Similarly, PG supplementation in animal models suggests that the ergogenic effects could be attributed to the changes in cellular bioenergetics and lipid metabolism. For example, increased swimming time duration, which was attributed to the stimulation of mitochondrial biogenesis in the myoblast has been described after supplementation with PG [24]. Endurance training in rats, which consisted of 20–25 m/min running on a slope of 8° for one hour five times a week, demonstrated that PG supplementation for 2 weeks increased fat oxidation and induced a glycogen-sparing effect in muscle. Likewise, in humans, metabolomic analysis revealed that ginseng induces changes in metabolic profiling primarily involved in lipid metabolism, energy balance, and chemical signaling in scull rowers. It was suggested that the change in metabolic profile could reflect the antifatigue effect of ginseng [12]. Similarly, volunteers treated with Panax ginseng extract showed an increase in carbohydrate and protein intake before the 10 km race and sub-maximal aerobic test, respectively. There is not enough information in the literature to explain this change in dietary behavior, although some authors suggest that ginsenosides may modify the behavior of neuropeptides related to satiety [25].

The ex vivo analysis performed in the present study confirms that PG may induce changes in cellular energy metabolism. In this regard, there was an observed reduction in oxygen consumption, and an increase in spare respiratory capacity in L6 cell culture exposed to PG volunteers’ serum. The increase in spare respiratory capacity suggests that cells would have extra ATP production by oxidative phosphorylation in case of a sudden increase in energy demand [26]. Increased spare respiratory capacity induced by PG extracts has been described in the H9C2 cell line (cardiomyoblast) after 12 h of exposure [27]. Since no changes in respiratory exchange ratio during sub-maximal exercise were observed in this study, it could be proposed that PG induces changes in bioenergetics towards higher energy efficiency. For further reading, a brief review of the literature describes the bioenergetics effects Rb1 (one of the most abundant ginsenosides from the Panax genus) regulating mitochondrial energy metabolism, fission and fusion, apoptosis, oxidative stress, reactive oxygen release, mitophagy, and mitochondrial membrane potential [28]. Notwithstanding, an increase in lactate levels was also observed (Table 4). This finding may suggest that the lower oxygen consumption observed in the cell culture could be compensated by an increase in anaerobic metabolism. Similarly, a significant increase in lactate levels was observed in volunteers supplemented with PG extract during the 10 km race. This increase was not significantly observed in volunteers supplemented with PG extract during the sub-maximal aerobic exercise test. One possible explanation for this observation is that the effort exerted in the 10 km race may have been greater than that in the laboratory sub-maximal aerobic exercise test. In fact, in both groups, perceived exertion values measured by the Borg’s scale were higher in the 10 km race than in the sub-maximal aerobic exercise test. All of this could confirm the initial hypothesis of the study that the ergogenic aid of PG may be more evident in sports activities performed at sub-maximal aerobic capacities.

Higher energy efficiency is also associated with a reduction in muscle damage after exercise. Ginseng supplementation in mice produced a reduction in blood creatine kinase in mice after endurance swimming [29], nitrate concentrations, and carbonyl content [7], which could be attributed to lower muscle damage. The observed changes in cytokine profile in the present study may reinforce this hypothesis (Table 2). For example, Rg1 ginsenoside supplementation in humans enhanced the exercise-induced IL-6 response [30], while others have observed enhanced exercise-induced IL-10 response [31]. In both studies, it is suggested that ginsenosides may improve immune cell infiltration in muscle tissue, which may improve senescent cell clearance [30] and increase muscle fiber renewal [31].

This study has some limitations. Although a reduction in oxygen consumption, blood lipid levels, and higher blood cytokine after exercise was observed due to PG supplementation, the molecular mechanisms could not be elucidated. To deepen this aspect, an ex vivo study was carried out in L6 muscle cells. However, the contraction of these cells was not stimulated, so the observed changes in metabolism can only be attributed to resting conditions. Additionally, the information collected in this study was obtained from a small number of volunteers and only from male subjects. Further studies should be conducted to validate these results in a larger cohort and female populations. Finally, changes in dietary behavior should also be considered in further study to exclude its effects on lipid homeostasis.

## 5. Conclusions

It was concluded that PG extract supplementation with higher doses of ginsenosides induces a reduction in oxygen consumption during the sub-maximal aerobic test. This observation was accompanied by a reduction in fasting TG levels. Thus, it is suggested that the ergogenic effects of PG supplementation may be related to greater efficiency in energy metabolism.

## Figures and Tables

**Figure 1 biomolecules-14-00533-f001:**
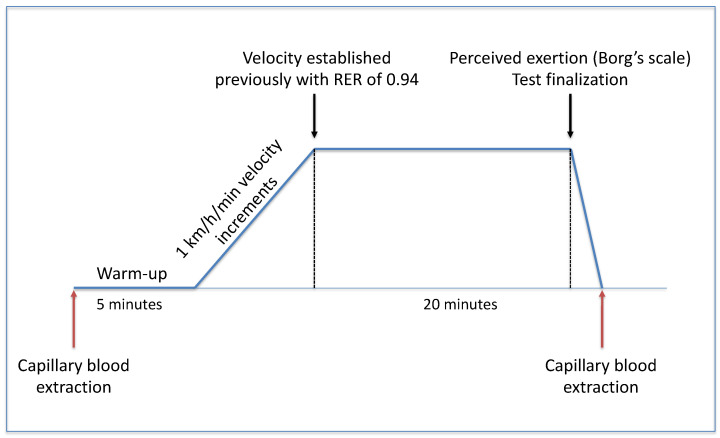
Workflow of the sub-maximal aerobic exercise test.

**Figure 2 biomolecules-14-00533-f002:**
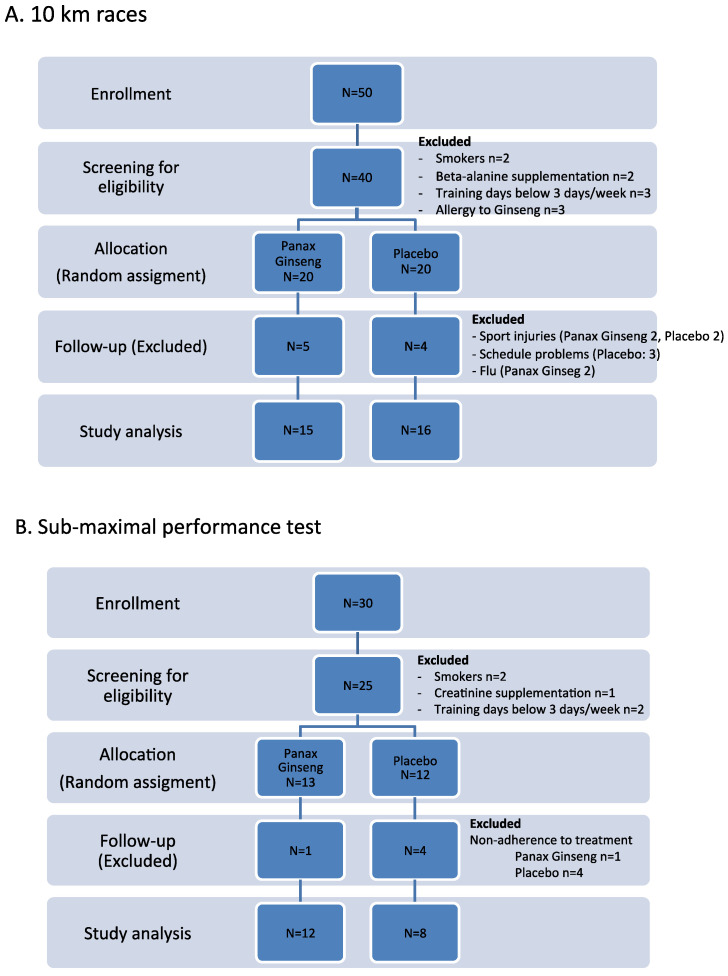
Recruitment flow chart.

**Figure 3 biomolecules-14-00533-f003:**
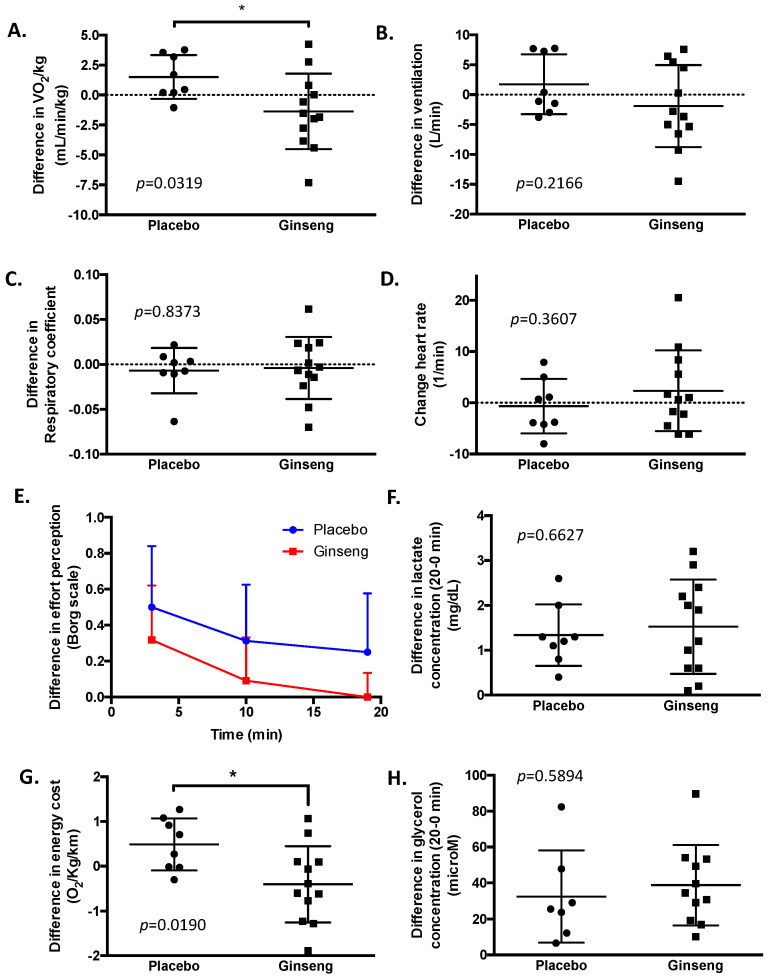
The effect of PG extract supplementation on changes in physical performance during the sub-maximal exercise test. The graphs describe the differences in physical performance parameters at time 0 and after 2 weeks of supplementation. Significantly (* *p* < 0.05) lower levels of VO_2_/kg of body weight were observed in the volunteers treated with PG (**A**), while no differences were observed in ventilation (**B**), respiratory coefficient (**C**), heart rate (**D**), or effort perception determined by the Borg’s scale (**E**). Regarding blood parameters, no differences in plasma lactate and glycerol levels during the sub-maximal exercise test after 2 weeks of PG supplementation are described in (**F**,**H**). Finally, a reduction in energy cost (**G**) during the sub-maximal test was observed in the PG group.

**Table 1 biomolecules-14-00533-t001:** Volunteers’ training and nutritional characteristics. Data are presented as mean (95% CI of the mean (min-max)). A paired *t*-test was performed to determine statistical differences between pre- and post-race values within groups. *p*-Values < 0.05 are highlighted as ^a^. Two-way ANOVA was performed to determine statistical differences between treatment groups in the Appendix A.

	Placebo Group		Panax Ginseng Group	
	Basal	Two Weeks	Basal	Two Weeks
**A. 10 km races**				
Volunteers	*n* = 16		*n* = 15	
Age	37.8 (32.9–42.7)		34.3 (29.8–38.8)	
Body weight (kg)	75.7 (71.6–79.8)	74.6 (70.0–79.3)	74.5 (70.1–78.9)	74.6 (70.0–79.3)
*Training parameters*				
Km/week	21.7 (12.8–23.8)	34.8 (25.4–44.2)	26.5 (19.5–33.4)	34.8 (28.4–41.1)
Min/week	113 (66–124)	192 (144–239)	135 (108–162)	183 (147–219)
*Nutritional parameters before race*				
Energy (kcal)	491 (380–602)	429 (339–519)	427 (318–535)	537 (373–700)
Carbohydrates (g/kg body weight)	0.81 (0.61–1.0)	0.73 (0.62–0.84)	0.75 (0.54–0.95)	**0.97 (0.76–1.19) ^a^**
Protein (g)	18.8 (12.4–25.3)	15.9 (11.5–20.4)	16.5 (10.8–22.2)	19.8 (12.7–26.9)
Lipids (g)	15.2 (10.7–19.7)	14.5 (9.2–19.9)	12.7 (8.1–17.4)	15.6 (5.2–25.9)
Hydration (mL)	486 (344–628)	481 (267–695)	598 (404–792)	532 (368–698)
**B. Sub-maximal performance test**				
Volunteers	*n* = 8		*n* = 12	
Age	37.4 (32.3–42.6)		34.2 (27.6–40.7)	
Body weight (kg)	72.7 (68.0–77.4)	72.8 (68.5–77.2)	71.1 (66.4–75.8)	71.6 (67.2–76.0)
*Training parameters*				
Km/week	30.1 (16.1–44.2)	24.1 (13.6–34.7)	32.5 (21.5–43.6)	33.7 (15.1–52.5)
Min/week	201 (109–292)	151 (94–208)	167 (103–231)	183 (102–264)
*Nutritional parameters before test*				
Energy (kcal)	646 (422–869)	674 (458–889)	513 (402–624)	669 (478–860)
Carbohydrates (g/kg body weight)	1.18 (0.60–1.75)	1.18 (0.80–1.56)	0.96 (0.70–1.21)	**1.13 (0.79–1.47) ^a^**
Protein (g)	23.9 (13.5–34.3)	22.7 (16.6–28.9)	17.7 (13.5–21.9)	**24.4 (16.9–31.8) ^a^**
Lipids (g)	21.3 (13.2–29.5)	23.9 (11.0–36.7)	17.3 (12.3–22.3)	25.4 (15.8–35.0)

**Table 2 biomolecules-14-00533-t002:** Blood variables of athletes at 10 km races. Volunteers completed two 10 km races, with 2 weeks between them. Values are presented as mean (lower-upper 95% CI). A paired *t*-test was performed to determine statistical differences between pre- and post-race values within groups. *p*-Values < 0.05 are highlighted as ^a^. Two-way ANOVA was performed to determine statistical differences between treatment groups. *p*-Values < 0.05 for differences between treatments are highlighted as ^b^.

Parameter	Placebo Group (*n* = 16)		Panax Ginseng Group (*n* = 15)	
	Pre-Race	Post-Race	Pre-Race	Post-Race
**First race**				
Running time (min)		47.4 (43.8–51.1)		44.3 (41.4–47.1)
Borg’s perceived exertion (CR-10)		6.4 (5.2–7.5)		6.6 (5.5–7.8)
Blood lipid profile (mg/dL)				
Total lipids	832 (783–880)	919 (862–975)	824 (755–894)	933 (868–998)
Phospholipids	144 (127–161)	148 (134 161)	168 (140–195)	159 (130–188)
Total cholesterol	170 (153–187)	166 (146–187)	162 (139–185)	157 (138–175)
Triacylglycerides	135 (108–163)	164 (107–222)	101 (72–130)	108 (86–130)
Non-esterified fatty acids	1.93 (1.37–2.50)	1.70 (1.09–2.30)	1.25 (1.01–2.27)	1.26 (0.78–1.73)
Lactate (mmol/L)	1.56 (1.16–1.96)	**3.15 (2.61–3.70) ^a^**	1.50 (1.14–1.86)	**5.59 (4.12–7.06) ^a,b^**
Cytokines (pg/mL)				
IL-1Ra	211 (114–309)	249 (170–239)	340 (0–690)	367 (44–690)
IL-6	13.6 (4.4–22.8)	13.8 (4.5–23.0)	3.7 (0–8.8)	8.9 (1.7–16.1)
IL-8	10.2 (7.3–13.0)	13.3 (10.6–16.0)	8.1 (4.0–12.2)	10.7 (6.5–14.9)
IL-10	16.3 (0–37.5)	83.3 (39.3–127.3)	9.2 (0.7–17.6)	**43.1 (26.3–60.0) ^a^**
TNFα	15.6 (11.0–20.2)	18.1 (13.7–22.4)	13.2 (10.2–16.2)	15.6 (12.1–19.1)
**Second race**				
Running time (min)		45.5 (42.0–49.1)		42.8 (39.8–45.8)
Borg’s perceived exertion (CR-10)		6.9 (6.0–7.7)		7.4 (6.7–8.2)
Blood lipid profile (mg/dL)				
Total lipids	874 (813–935)	943 (881–1006)	**774 (711–837) ^a^**	959 (867–1051)
Phospholipids	153 (137–170)	152 (121–183)	150 (121–178)	152 (118–187)
Total cholesterol	180 (159–200)	172 (151–193)	155 (133–176)	162 (145–179)
Triacylglycerides	138 (103–174)	126 (103–148)	**93 (72–114) ^a,b^**	95 (75–115)
Non-esterified fatty acids	1.02 (0.94–1.56)	1.70 (1.24–2.14)	0.51 (0.73–1.81)	1.45 (1.04–1.86)
Lactate (mmol/L)	1.33 (1.12–1.53)	**3.04 (2.37–3.71) ^a^**	1.26 (1.07–1.45)	**4.85 (3.03–6.67) ^a,b^**
Cytokines (pg/mL)				
IL-1Ra	275 (106–443)	271 (134–409)	231 (73–389)	333 (0–682)
IL-6	7.4 (3.7–11.0)	10.0 (5.1–14.9)	**5.1 (0–10.4)**	**7.2 (3.5–10.9) ^a^**
IL-8	12.4 (7.8–17.0)	13.9 (11.0–16.8)	8.7 (4.4–13.1)	**12.9 (9.0–16.8) ^a^**
IL-10	11.6 (2.9–20.3)	**86.7 (39.7–133.6) ^a^**	8.2 (2.6–13.9)	**76.2 (37.6–114.8) ^a,b^**
TNFα	16.9 (13.4–20.4)	18.8 (14.9–22.8)	13.5 (11.1–15.9)	16.4 (13.5–19.4)

**Table 3 biomolecules-14-00533-t003:** Blood and aerobic capacity variables of athletes during the sub-maximal performance test. Volunteers completed two sub-maximal performance tests within 2 weeks between them. Aerobic capacity variables are the mean of the values recorded during the 20 min sub-maximal aerobic test. Values are presented as mean (lower-upper 95% CI). A paired *t*-test was performed to determine statistical differences between pre- and post-test values within groups. *p*-Values < 0.05 are highlighted as ^a^. Two-way ANOVA was performed to determine statistical differences between treatment groups. *p*-Values < 0.05 for differences between treatments are highlighted as ^b^.

Parameter	Placebo Group (*n* = 8)		Panax Ginseng Group (*n* = 12)	
	Pre-Test	Post-Test	Pre-Test	Post-Test
**First test**				
VO_2_ (mL/min/kg)		36.0 (33.5–38.5)		41.2 (35.4–47.1)
VCO_2_ (mL/min/kg)		34.1 (31.6–36.6)		39.0 (33.6–44.3)
RER		0.95 (0.93–0.96)		0.95 (0.93–0.95)
VE (L/min)		29.1 (26.3–31.9)		29.2 (27.5–30.9)
VE/VO_2_		27.7 (25.0–30.3)		27.7 (26.1–29.4)
Heart rate (bpm)		154 (148–161)		152 (143–161)
Borg’s perceived exertion (CR-10)		4.4 (2.9–5.9)		3.9 (3.2–4.6)
Lactate (mmol/L)	2.1 (1.8–2.5)	**4.0 (3.3–4.7) ^a^**	2.3 (2.7–3.9)	**3.6 (2.9–4.3) ^a^**
Blood lipid profile (mg/dL)				
Total lipids	599 (556–642)	665 (557–772)	671 (593–749)	679 (605–754)
Phospholipids	184 (163–204)	210 (178–242)	175 (156–194)	202 (173–232)
Total cholesterol	183 (134–231)	182 (133–231)	165 (138–191)	173 (148–197)
Triacylglycerides	63 (48–79)	71 (58–84)	85 (56–114)	93 (67–120)
Non-esterified fatty acids	0.96 (0–2.32)	1.03 (0.46–1.60)	0.67 (0.29–1.06)	1.36 (0.62–2.10)
Cytokines (pg/mL)				
IL-1Ra	168 (31–305)	214 (84–344)	114 (42–186)	132 (76–189)
IL-6	4.1 (1.2–7.1)	6.7 (2.5–10.9)	16.8 (0.0–42.8)	23.1 (0.0–63.3)
IL-8	8.5 (4.2–12.8)	12.2 (4.6–19.8)	7.0 (4.2–9.8)	7.9 (4.9–11.0)
IL-10	4.2 (2.1–6.4)	**6.9 (3.5–10.4) ^a^**	8.0 (3.8–12.1)	9.6 (3.7–15.5)
TNF-α	14.3 (11.6–17.0)	**21.8 (15.2–28.5) ^a^**	13.5 (9.4–17.7)	14.5 (10.2–18.8)
**Second test**				
VO_2_ (mL/min/kg)		37.8 (34.5–41.1)		39.8 (34.1–45.5)
VCO_2_ (mL/min/kg)		35.6 (32.7–38.5)		38.2 (33.1–43.3)
RER		0.94 (0.92–0.96)		0.95 (0.92–0.96)
VE (L/min)		29.1 (26.8–31.3)		29.7 (28.1–31.4)
VE/VO_2_		27.4 (25.1–29.7)		28.1 (26.3–30.0)
Heart rate (bpm)		154 (148–161)		155 (146–164)
Borg’s perceived exertion (CR-10)		4.4 (3.2–5.8)		4.0 (3.1–5.0)
Lactate (mmol/L)	2.0 (1.5–2.5)	**3.3 (2.7–4.0) ^a^**	2.1 (1.7–2.5)	**3.6 (2.9–4.3) ^a^**
Blood lipid profile (mg/dL)				
Total lipids	660 (586–735)	688 (650–727)	**592 (519–665) ^a^**	670 (599–741)
Phospholipids	184 (165–204)	215 (189–241)	174 (151–197)	204 (175–233)
Total cholesterol	189 (143–234)	217 (164–270)	165 (143–186)	181 (153–209)
Triacylglycerides	65 (57–74)	69 (55–83)	**71 (52–90) ^a^**	86 (60–113)
Non-esterified fatty acids	1.57 (0–3.58)	0.97 (0.07–1.87)	0.65 (0.45–0.84)	1.46 (0.88–2.05)
Cytokines (pg/mL)				
IL-1Ra	167 (95–239)	255 (88–422)	130 (35–226)	**175 (47–303) ^a,b^**
IL-6	4.0 (1.5–6.4)	5.2 (0.3–10.1)	2.6 (0.4–5.7)	2.6 (0.3–4.9)
IL-8	8.5 (2.9–14.1)	10.7 (5.5–15.9)	6.7 (4.5–8.8)	8.0 (5.2–10.8)
IL-10	5.9 (2.2–9.8)	8.6 (4.2–13.0)	7.7 (2.2–13.2)	**13.6 (0–28.2) ^a^**
TNF-α	13.8 (9.4–18.3)	18.3 (10.3–26.3)	14.4 (10.9–17.9)	19.2 (9.1–29.2)

**Table 4 biomolecules-14-00533-t004:** L6 cell culture respiratory parameters after 24 h treatment with volunteers’ serum. Values are presented as mean ± SEM of pmol/s per mg of protein. Lactate was measured in cell lysis supernatant after the respiratory protocol. Differences in lactate content were observed between placebo and ginseng-treated cells (*p* = 0.0375) *.

Parameter	Placebo Group			Panax Ginseng Group		
	Pre-Treatment	Post-Treatment	*p*-Value	Pre-Treatment	Post-Treatment	*p*-Value
Routine respiration	292 ± 53	304 ± 23	0.8435	383 ± 51	232 ± 31	0.0639
Uncoupled respiration	525 ± 110	651 ± 113	0.4725	585 ± 66	607 ± 50	0.8098
Spare respiratory capacity	234 ± 57	348 ± 95	0.3627	202 ± 63	374 ± 20	0.0592
Lactate (mg/dL)		2.04 ± 0.22			2.84 ± 0.26 *	

## Data Availability

The data presented in this study are available on request from the corresponding author.

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
