# Peer review of "Short-Term Panax Ginseng Extract Supplementation Reduces Fasting Blood Triacylglycerides and Oxygen Consumption during Sub-Maximal Aerobic Exercise in Male Recreational Athletes"

_biomolecules, 2024, doi:10.3390/biom14050533_

Round 1
Reviewer 1 Report
Comments and Suggestions for Authors
I have some specifc comments for Authors, I hope they will be helpful:
L20 2.9 L/min/kg - such a change is impossible!
L48 VO2max: please explain the abbreviation first and then use it consistently
L64 please justify the duration of supplementation in the introduction
L72-79 Did the same subjects participate in both stages (experiments), despite different sample sizes? If so, was the washout period taken into account? this part needs to be clarified and detailed.
L82: were both runs performed under the same environmental conditions? Please specify the time of blood collection and Borg scale - 'before and after' is too general. The methodology should be detailed enough to be replicated.
L90 "submaximal aerobic capacity" - i do not know such a term in exercise physiology. i think it meant submaximal aerobic exercise or submaximal aerobic test. Please consult the terminology
section 2.1.2 is unclear. was the test performed up to RER=0.94 and then stopped? How were AT and AnT determined? Was VO2max vs. VO2peak verified? How was the warm-up performed? Why was RER=0.94 determined how were metabolic thresholds determined at which metabolism is known?
How was the sample size calculated?
Did the participants train during the intervention? If so, you need to describe the training.
Was the diet of the participants analyzed? without analyzing the diet it is impossible to draw valid conclusions
Please report the effect size
ANOVA: please report results (f, p, partial eta squared) for main effects
Author Response
Reviewer No. 1
I have some specifc comments for Authors, I hope they will be helpful:
Thanks for the comments, certainly they are very helpful. Some typographic errors were included in the initial manuscript that we did not observe. Similarly, we have included all the recommendations raised to increase the clarity of the text.The results of the statistical analyses were included as supplementary information. We have implemented all suggested changes highlighted in red text within the article. In blue text, you will find the suggestions from Reviewer No. 2.
Answers to the comments
L20 2.9 L/min/kg - such a change is impossible!
This was a typographic error, the change in VO2 was 1.4 mL/min/kg.
L48 VO2max: please explain the abbreviation first and then use it consistently
“Maximum Oxygen Consumption (VO2max) was added to the text
L64 please justify the duration of supplementation in the introduction
The rationale for the dosage and exposure time was included in the 2.3 Intervention section. Basically, it was proposed that since previous studies had not observed any effects on physical activity, it was decided to try a higher dose. On the other hand, to avoid potential adverse effects, a short exposure time was chosen.
L72-79 Did the same subjects participate in both stages (experiments), despite different sample sizes? If so, was the washout period taken into account? this part needs to be clarified and detailed.
There were different groups of volunteers in each trial. We have included in the text, that different volunteers in each group were included. This aspect may reinforce our findings since the total number of volunteers that take Panax Ginseng was 28.
L82: were both runs performed under the same environmental conditions? Please specify the time of blood collection and Borg scale - 'before and after' is too general. The methodology should be detailed enough to be replicated.
Both runs were performed under the same environmental conditions. The perceived fatigue was assessed using the Borg scale immediately after the volunteer crossed the finish line. Part of the research team directed the volunteer to the station that the research group had near the finish line, and capillary blood was obtained immediately within a maximum time of 5 minutes. We have changed the text as follows:
“ Blood samples were obtained one day before the race in fasting conditions and after each race immediately after the volunteer crossed the finish line. Part of the research team directed the volunteer to the station that the research group had near the finish line, and capillary blood was obtained immediately within a maximum time of 5 minutes. Organization electronic chips recorded the race time, and the general feeling of perceived fatigue was determined by Borg’s rating of perceived exertion CR-10 scale immediately after the end of both races.”
L90 "submaximal aerobic capacity" - i do not know such a term in exercise physiology. i think it meant submaximal aerobic exercise or submaximal aerobic test. Please consult the terminology
We agree with the reviewer that “capacity” is not an adequate terminology for the test performed. We have changed the title as “Submaximal aerobic test” and the definition with “Submaximal aerobic exercise”.
section 2.1.2 is unclear. was the test performed up to RER=0.94 and then stopped? How were AT and AnT determined? Was VO2max vs. VO2peak verified? How was the warm-up performed? Why was RER=0.94 determined how were metabolic thresholds determined at which metabolism is known?
The protocol that was carried out was as follows: First, a maximal aerobic capacity test was performed to determine the aerobic threshold. Based on these results, the speed at which the volunteer had an RER of 0.94 was estimated. This speed was to be maintained in the second session, which would be set and define as sub-maximal aerobic exercise. The second session involved a 5-minute warm-up at 6 km/h, followed by speed increments of 1 km/h every minute until the volunteer reached the speed established in the first test (equivalent to an RER of 0.94). Once this speed was reached, it was maintained constant for 20 minutes. The RER during these 20 minutes showed slight variations (included in Table 3); which allowed us to verify that the volunteer performed sub-maximal aerobic exercise for the 20 minutes. To clarify the protocol, we have included a figure outlining the protocol used.
How was the sample size calculated?
The information regarding the sample size calculation is included in the section 2.2. We have noticed that the minimum number of participants per group in each experiment was not included in the text. Therefore, we have added this information.
Did the participants train during the intervention? If so, you need to describe the training.
Was the diet of the participants analyzed? without analyzing the diet it is impossible to draw valid conclusions
The study participants continued their usual training during the experimental period. This information is summarized in Table 1. Additionally, we have included in Table 1 the information on nutritional intake before the test. There is an increase in carbohydrate and protein intake in the Panax Ginseng-supplemented group. This change in dietary intake is difficult to explain with the variables collected in the study. Nonetheless, we have included a section in the Results and Discussion where these findings are described. Similarly, we have included a reference to a study suggesting that ginsenosides could affect satiety parameters. Additionally, in the study limitations section, we have noted that Panax ginseng could affect dietary behavior, which in turn may affect some of the variables collected in this study.
Please report the effect size. ANOVA: please report results (f, p, partial eta squared) for main effects
This information is included as a supplementary file.
Reviewer 2 Report
Comments and Suggestions for Authors
The manuscript entitled “Panax Ginseng supplementation reduces fasting blood triacylglycerides and oxygen consumption during sub-maximal aerobic capacity in male recreational athletes” presents the evaluation short-term effects of Panax ginseng extract administration on on aerobic capacity, lipid profile, and cytokines in a 14-day randomized, double-blind trial. The manuscript is quite interesting and well-written. However, major revisions should be made in order to be published in Biomolecules journal, and the manuscript should be completed and/or modified taking into account the suggestions from the attached file.

Comments on the Quality of English Language
Minor editing of English language is required
Author Response
Reviewer No. 2
We appreciate the feedback from the review, which has helped us improve the information presented in the article. We have implemented all suggested changes highlighted in blue text within the article. In red text, you will find the suggestions from Reviewer No. 1. In summary, we have included the word 'extract' as it is indeed accurate that there is a difference between the consumption of the plant itself and an ethanolic extract, which was used in this study. We have also provided a more extensive discussion of the lactate results in this study. Overall, we did not observe any significant changes in the values obtained from the volunteers. Significant changes were only observed in the case of the cell culture, which could indicate that low oxygen consumption may lead to an increase in glycolysis. We believe that the effects of Panax Ginseng could impact mitochondrial metabolism; however, as we did not observe the same effect in the volunteers, it is difficult for us to extrapolate this observation to a whole organism.
The manuscript entitled “Panax Ginseng supplementation reduces fasting blood triacylglycerides and oxygen consumption during sub-maximal aerobic capacity in male recreational athletes” presents the evaluation short-term effects of Panax ginseng extract administration on on aerobic capacity, lipid profile, and cytokines in a 14-day randomized, double-blind trial.
The manuscript is quite interesting and well-written. However, major revisions should be made in order to be published in Biomolecules journal, and the manuscript should be completed and/or modified taking into account the suggestions below:
- The authors are advised to change the title of the manuscript, in order to use the term „extract”. Since the authors used an standardized extract of Panax ginseng root and write about the importance of choosing the right dose for the efficacy, it should be mentioned.
The title was changed following the reviewer suggestion. The dose and time of exposure was clarified in the methodology section 2.3 Interventions.
- Line 11: the authors are advised to use the term „extract” (Panax ginseng extract)
The term “extract” was added.
- The authors are advised to rephrase the sentences from lines 30-33, 52-53, 207-214.
The cited sentences were rephrased.
- In 2.3. section, the authors are advised to explain how they choose the doses (500mg extract). Furthermore, they should better explain if they made themselves the extract, or it was purchased, also from ACEF
The extract was purchased directly from ACEF.
- For 2.4.3. subsection, the authors are advised to add references
We add a reference of a general procedure in cell respirometry using Oroboros equipment.
- In Results section, the authors are advised to also explain the values from lactate level, for both Placebo and Ginseng groups, for the two races. The same for results from table 6.
We add an explanation of lactate levels in both results and discussion section. We hope that the added information will clarify some aspects of the presented findings.
- The authors are advised to use the same names (as presented in section 2) for the groups: placebo group and ginseng group, for all tables, for better understanding.
The titles used for each group in the tables were standardized.
- The authors are advised to better explain the phrase from lines 271-272. How they validated the obtained results by using the L6 muscle cell lines? They also should better explain why an increase in cellular lavctate levels was observed?
We agree with the reviewer that the term “validate” is not correct. We do not pretend to validate human findings with in vitro results. We have eliminated the phrase “validate” to avoid misunderstandings to the readers.
- The authors are advised to develop the phrase from lines 320-321, since they were the authors of the study from reference 19.
We have included the specific results observed from the meta-analysis, mainly a reduction in total and LDL-cholesterol levels.
- Also, they are advised to complete the discussion section about the lactate levels
A discussion regarding lactate results was added.
- Among the limitations of the study, one can notice the small number of subjects.
We have included the small number of subjects as a limitation of the study.
Round 2
Reviewer 1 Report
Comments and Suggestions for Authors
The Authors significantly improved the manuscript. Thank you.
Minor comment:
L68 please add "performance" i.e. "on submaximal aerobic exercise performance". If necessary, change elsewhere in the manuscript as well.
Author Response
Thank you for your comments. We have included the term "performance" in line 68, 80 and 286 following your suggestions.
We appreciate your time and expertise devoted to reviewing the article.
Reviewer 2 Report
Comments and Suggestions for Authors
The authors made all the required changes and the manuscript has been significantly improved.
Comments on the Quality of English Language
Minor editing of English language is required
Author Response
We want to thanks again for your time and expertise devoted to reviewing the article. The manuscript was reviewed again focusing on the spelling and grammar aspects. The included changes are highlighted in green.